# Sex differences in human gregariousness

Joyce F. Benenson, Sandra Stella and Anthony Ferranti

Emmanuel College, Boston, MA, USA

## ABSTRACT

Research on human sociality rarely includes kinship, social structure, sex, and familiarity, even though these variables influence sociality in non-human primates. However, cross-cultural ethnographic and observational studies with humans indicate that, beginning after age 5 years, males and females form differing social structures with unrelated individuals in a community. Specifically, compared with females, human males exhibit greater tolerance for and form larger, interconnected groups of peers which we term "gregariousness." To examine sex differences in gregariousness early in life when children first interact with peers without adult supervision, 3- to 6-year-old children were given the choice to enter one of three play areas: an empty one, one with an adult, or one with a familiar, same-sex peer. More males than females initially chose the play area with the same-sex peer, especially after age 5 years. Sex differences in gregariousness with same-sex peers likely constitute one facet of human sociality.

## INTRODUCTION

Sex differences in sociality in non-human mammals are well-documented. With few exceptions, mammalian females spend more time than males with offspring (*Trivers, 1972*), whereas interaction with unrelated same-sex individuals varies by species (*Wrangham, 1987*). In humans' two closest living genetic relatives, chimpanzees (*Pan troglodytes*) and bonobos (*Pan paniscus*), sex differences in interaction with unfamiliar same-sex individuals vary. Male chimpanzees interact in groups of related and unrelated males, whereas females are more solitary, typically interacting with their offspring or one or two other unrelated adult females (*De Waal, 1982/2007*; *Goodall, 1986*; *Langergraber, Mitani & Vigilant, 2009*; *Lehmann & Boesch, 2009*; *Wrangham, 2000*). In contrast, female bonobos interact in groups of unrelated females along with their offspring, whereas males are relatively solitary (*Chapman, White & Wrangham, 1994*; *Kanō, 1992*; *White & Chapman, 1994*).

In humans, mothers likewise invest more than fathers in their children, especially infants and vulnerable children (*Brown, 1991*; *Konner, 2005*; *Konner, 2010*). Researchers extrapolate this finding to all classes of individuals, producing the conclusion that human females are more sociable overall than males (*Taylor et al., 2000*; *Winstead & Griffin, 2001*). Narrower definitions of sociality rarely occur.

Corresponding author
Joyce F. Benenson,
Joyce.Benenson@gmail.com

Observational studies however suggest human males exhibit more gregariousness than females toward unrelated same-sex peers early in childhood. By 6 months, male infants look longer than female infants at groups (*Benenson, Duggan & Markovits, 2004*). By 3 years, males interact with more same-sex peers than females (*Fabes, Martin & Hanish, 2003*). After 5 years, cross-culturally males more than females spontaneously form organized groups of same-sex peers (*Barbu, Cabanes & Le Maner-Idrissi, 2011*; *Benenson, Apostoleris & Parnass, 1997*; *Fine, 1980*; *Savin-Williams, 1980*) and interact more frequently with same-sex peers versus kin throughout middle childhood, adolescence and adulthood (*Benenson et al., 2008*; *Schlegel & BarryIII, 1991*; *Whiting & Edwards, 1988*).

Additional reports indicate that males exhibit greater tolerance for transgressions of same-sex friends in middle childhood (*MacEvoy & Asher, 2012*), adolescence (*Whitesell & Harter, 1996*), and adulthood (*Benenson et al., 2009*). Accordingly, same-sex peer relationships of males endure longer than females in middle childhood and adolescence (*Benenson & Alavi, 2004*; *Benenson & Christakos, 2003*; *Kon & Losenkov, 1978*; *Montemayor & Van Komen, 1985*) and young adulthood (*Benenson et al., 2009*).

However, no pure experimental test of sex differences in gregarious with same-sex peers exists. Recently, we attempted to test sex differences in children's sociability with adults versus peers by creating three play areas and placing a randomly chosen same-sex peer in one of them, a female adult in a second, and leaving the third area empty (*Benenson, Quinn & Stella, 2012*). A focal child then was given the choice of where to play. Across eight minutes, no sex differences occurred in total time spent with the same-sex peer, but more males than females spent at least 30 s with the peer and males entered and exited the peer's play area more frequently.

This study however is open to several criticisms. First, the peer influenced the focal child's decisions. Second, including only a female adult as the alternative social partner may have biased the results by providing two same-sex social partners for female children but only one for male children. Consequently, we decided to re-analyze the results by examining only the focal child's first choice of play area. Additionally, we conducted a second study with a male instead of a female adult and again examined only the child's first choice of play area. We then merged the two data sets. Our test of gregariousness became willingness to initiate interaction with a randomly chosen, familiar same-sex unrelated child versus with an adult. We wanted to include children as young as possible to minimize the strength of socialization influences. Thus, we included children when they first begin to interact with peers without adult supervision in early childhood beginning at 3 years of age. We also wanted to test whether the formation of interconnected groups which emerges spontaneously in boys around 5 years would accentuate any sex differences. We hypothesized that more males than females would choose the play area containing the same-sex peer, especially after age 5 years when organized group interaction emerges.

## MATERIALS AND METHOD

The Committee for the Protection of Human Participants in Research (CPHPR) of Emmanuel College reviewed and approved this research, protocol # Benenson_08.15.12.

Combining the data from the original study (*Benenson, Quinn & Stella, 2012*) and the new study yielded 84 pairs of primarily Caucasian children between 37 and 72 months ($M_{age}$ = 56.53 months, $SD$ = 10.84, $n$ = 36) and ($M_{age}$ = 56.71 months, $SD$ = 10.60, $n$ = 48) from 4 schools in lower- to upper-middle socioeconomic (SES) neighborhoods in Boston, Massachusetts. Each pair contained a focal child and a second child of the same sex from the same classroom.

The procedure in the second study was identical to the original study (*Benenson, Quinn & Stella, 2012*) with the only difference being a male instead of a female adult occupied one of the three play spaces. For both studies, children with written parental permission to participate within every classroom were randomly paired with another child of the same sex. One was then randomly assigned to be the focal child. Verbal assent was obtained individually from each child before beginning the procedure. Only when both children agreed to participate was the pair included in the study.

In a room at each school, three identical circular play spaces were erected in an equilateral triangle such that each enclosure was 101.6 cm (40 in) at its closest point to the triangle's epicenter. Each play area was enclosed by free-standing fiberglass, 121.9 cm (4 ft) high and 508 cm in circumference with a 76.2 cm opening (door) that was invisible to the focal child who was placed at epicenter of the three spaces. In each play area were identical boxes containing children's play materials arrayed in identical positions vis-à-vis the opening to the house. The height of the play spaces prevented visual communication between children. To prevent auditory communication, audiotapes of children's singing played continuously.

An unfamiliar adult (one of four females in the original study and one of two males in the second study) met the pair of children in their classroom, brought the pair to the room containing the three play areas, and established a rapport with the children along the way while reciting a standardized script describing the three "houses." Once the adult and pair of children arrived, the adult escorted one child to play in one of the play areas, then entered a second play area. A second adult (one of three female adults, two of whom participated in both the original and the second study) simultaneously greeted the second (focal) child and delivered the following script while showing the focal child each play area:

(Name of focal child), you get to choose wherever you want to play. You can go into any house you want, and you can change houses whenever you want. Each house has the exact same fun things in it. I am going to show you the door to each house. You can play with (name of peer) and the fun things in this house; You can play with (name of adult) and the fun things in this house; You can play with the fun things in this house.

The focal child then was led to the epicenter of the three play areas facing towards the door of the room and permitted to play. The director immediately moved toward the door and began circling the houses clockwise while recording the focal child's first choice of play area. The children then were permitted to play in the area they chose before returning to their classrooms. The order of the houses in terms of their occupant vis-à-vis the door and in the script was different in each school.

**Table 1** Percent of children who chose to play with the peer.

| Age (months) | Males | | Females | | $X^2_1$ sex | P |
| --- | --- | --- | --- | --- | --- | --- |
| | (%) | n | (%) | n | | |
| 37–59 | 54.5 | 12/22 | 37.5 | 9/24 | 1.34 | n.s. |
| 60–72 | 78.6 | 11/14 | 45.8 | 11/24 | 3.89 | .049 |
| Total | 63.9 | 23/36 | 41.7 | 20/48 | 4.07 | .044 |

## RESULTS

All focal children entered a play area within 5 s. We used a chi-square test to compare the total number of male versus female focal children across both studies who chose the play area containing the same-sex peer. Significantly more male than female children chose the play area containing the same-sex peer (see Table 1). The same analysis was then conducted with children younger than 5 years (37–59 months) and again with children over 5 years (60–72 months). As depicted in Table 1, at both ages more male than female focal children chose the play area containing the same-sex peer, but the sex difference was significant only at the older age.

The sex of the adult had no significant effect (female adult: 68% of the males vs. 50% of the females chose the same-sex peer; male adult: 59% of the males vs. 33% of the females chose the same-sex peer). Comparing the number of males who chose the same-sex peer with the male versus female adult produced a $X^2_1 = 1.48, p > .20$. Likewise, comparing the number of females who chose the same-sex peer with the male versus female adult yielded a $X^2_1 = 2.63, p > .10$. Finally, we examined sex differences in focal children's choice of the play space containing the adult. Too few children (one girl with the female adult and one boy and two girls with the male adult) selected to enter the play area containing the adult to permit analysis. Children who did not enter the play areas containing the peer or adult chose the empty play area.

## DISCUSSION

Results supported the hypothesis that in early childhood, males affiliate more than females with a randomly chosen, familiar same-sex peer. The simple procedure constitutes a relatively pure experimental test of sex differences in children's gregariousness with an unrelated, familiar same-sex peer. The results are consistent with past findings that compared with females, males exhibit greater tolerance for peers' transgressions and by age 5 years interact more in organized same-sex groups.

Theoretically, cooperation between human males may be more related to survival through cooperative defense and similar joint activities (*Bowles, 2009*). Empirical research likewise indicates that beginning at age 5 years and continuing into adulthood, unrelated human males are more likely than unrelated females to engage in cooperative activities (*Barbu, Cabanes & Le Maner-Idrissi, 2011*; *Benenson, Apostoleris & Parnass, 1997*; *Hall, 2011*; *Winstead & Griffin, 2001*). Females in turn provide more reciprocal verbal support (*Buhrmester & Prager, 1995*).

Gregariousness with a randomly chosen, familiar same-sex peer constitutes only one type of sociality. Other types of sociality, including close friendships, mixed-sex partnerships, kin relations, and alliances between unfamiliar individuals within and between communities, likely exhibit different patterns. Cross-culturally, girls typically remain in closer proximity than boys to kin (*Whiting & Edwards, 1988*). In societies where children attend school, girls spend as much time as boys in proximity to unrelated peers. Therefore, although girls may not be as driven as boys to initiate interaction with a same-sex peer, eventually they likely engage in similar quantities of interaction when separated from kin. Finally, had groups of same-sex peers been the focus, much evidence suggests that males would have invested more than females (*Benenson & Markovits, 2014*).

The significance of the sex difference in preference for same-sex peers at the older but not younger age may be due to the well-established pattern of increasing sex-segregation at this age (*Maccoby, 1998*). Alternatively, as boys in particular transition from interaction with adults to increased investment in same-sex peers (*Whiting & Edwards, 1988*), the sex difference may become more pronounced. Cross-culturally beginning at age 5 years, boys interact in organized same-sex groups (*Benenson, Apostoleris & Parnass, 1997*; *Benenson & Markovits, 2014*) and increased investment in individual same-sex peers may facilitate the transition from adult to peer group interaction. Future research is required to specify the developmental transitions that occur between reliance on adults to investment in same-sex peers.

The study is limited in that it included primarily Caucasian children from a range of middle SES backgrounds. Further, the number of pairs is small. Future studies that include a more diverse population with additional same-sex pairs of children are necessary to strengthen the validity of the findings.

## CONCLUSION

These results indicate that human males from a young age exhibit higher levels of gregariousness than females. Sex differences in gregariousness likewise appear in observations of 30–36 month-old chimpanzees who also are just beginning to play independently of their mothers (*Lonsdorf et al., 2014*). This is consistent with humans' and chimpanzees' similar social structures (*Rodseth et al., 1991*). More precise definitions and measures of sociality and inclusion of differing classes of individuals will further illuminate the nature of human sociality.

## ACKNOWLEDGEMENTS

We thank Ryan Lane and Victoria Lambert for their invaluable assistance with data collection, and the principals, teachers, parents, and children who made this study possible.

### Funding

The Emmanuel College Faculty Development Committee funded this research. The funders had no role in study design, data collection and analysis, decision to publish, or preparation of the manuscript.

### Competing Interests

The authors declare there are no competing interests.

### Author Contributions

- Joyce F. Benenson conceived and designed the experiments, performed the experiments, analyzed the data, contributed reagents/materials/analysis tools, wrote the paper, prepared figures and/or tables, reviewed drafts of the paper.
- Sandra Stella conceived and designed the experiments, performed the experiments, contributed reagents/materials/analysis tools, reviewed drafts of the paper.
- Anthony Ferranti performed the experiments, contributed reagents/materials/analysis tools, reviewed drafts of the paper.

### Human Ethics

The following information was supplied relating to ethical approvals (i.e., approving body and any reference numbers):

The Committee for the Protection of Human Participants in Research (CPHPR) of Emmanuel College reviewed and approved this research, protocol # Benenson_08.15.12.

### Supplemental Information

Supplemental information for this article can be found online at http://dx.doi.org/10.7717/peerj.974#supplemental-information.

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
