# Peer review of "Sex differences in human gregariousness"

_PeerJ, doi:10.7717/peerj.974_

## Round 0.1 · original submission · Minor Revisions

I have now received reviews from two experts regarding this submission. They were generally pleased with your manuscript, although both noted some limitations that I would like you to address in a revised submission.
The most important limitation is that the anonymous reviewer believes there should have been an opposite-sex peer play option. I suspect (and hope) that you can make a reasonable case (perhaps in the revised Discussion) as to why that option, although it would be advantageous to include, does not undermine your main conclusion.

·

Basic reporting

l.149: "who provide more reciprocal verbal support": Grammatically the referent of this phrase is "cooperative activities", which of course doesn't make sense. You need to change the sentence structure so that it actually refers to "unrelated human males".

Experimental design

The sample size is not very big - especially for the older age group - given that children are randomly paired so there is bound to be a lot of noise in the data. The paragraph on limitations (ll.159-161) is very short and should be expanded to include this point. Related to this point, you ought to include data on the total numbers of boys and girls in each class (apologies if I have missed this!). It could be a potential confound if there were many more of one sex than the other, as in the more numerous sex it would be more likely that a child would be paired with an unfamiliar or disliked same-sex peer, whom they would be more likely to avoid by playing alone.

Validity of the findings

l.132 "The sex of the adult had no effect": It's better to say that it had "no significant effect" and give the chi-squared and p values so that readers can see how far away from significance it was. For the girls, 50% to 33% is actually a bit of a jump so it does look possible that a larger sample size might reveal an effect there.

ll.151-152: "Gregariousness with a randomly chosen, familiar same-sex peer constitutes only one type of sociality": I would actually expect to see a much stronger sex difference with choosing groups of peers over lone peers as playmates. There is evidence that females more than males like strong dyadic or triadic emotional bonds, whereas males more than females like belonging to larger groups with a specific shared interest, e.g. Eder, D., & Hallinan, M. T. (1978). Sex differences in children's friendships. American Sociological Review, 43, 237-250; Maccoby, E. E. (2000). Perspectives on gender development. International Journal of Behavioral Development, 24, 398-406. You could mention this as a potential avenue for further research, using similar methods to those of the current study.

You don't really discuss the age difference (the fact that the sex difference was significant for older children but not for younger). This is in line with a lot of literature suggesting that voluntary gender segregation increases with age through middle childhood, e.g. Maccoby, E. E. (1998). The two sexes: Growing up apart, coming together. Cambridge, MA: Bellknap Press.

Additional comments

An interesting little study in an under-researched area. I look forward to seeing further work from your research group along these lines!

Reviewer 2 ·

Basic reporting

pass

Experimental design

pass

Validity of the findings

This is an interesting paper, and the authors make an important point that sociality/gregariousness among males vs. females has not been adequately parsed in the literature. Overall, the methods appear strong and the writing is good. My main concern is the manipulation of play options. Why did the authors not include a play area with an opposite-sex peer? Could they not have obtained their present results because of a greater interest in adults or solitary play among girls, and/or because of an aversion to adults and/or solitary play among boys? In other words, is it not the case that we can’t know whether boys have a preference for same-sex over opposite-sex playmates without having an opposite-sex (peer) playmate as an option? I can’t think of a good reason for not including an opposite-sex peer as an option (perhaps instead of an adult), but the authors should be given the opportunity to explain this decision. If their reasoning isn’t convincing, I would suggest re-running the study including the option of either a same-sex or an opposite-sex peer. Those results, plus the present ones, should clarify children’s preferences for same-sex playmates.

Additional comments

Minor Editing:
Line 32 “adult females” rather than “adult female”
Line 92 “triangle” instead of “triangular”

The authors may also find these additional references useful:
Oxford, J., Ponzi, D., & Geary, D. C. (2010). Hormonal responses differ when playing violent video games against and ingroup and an outgroup. Evolution and Human Behavior, 31, 201-209.
Rose, A. J., & Rudolph, K. D. (2006). A review of sex differences in peer relationship processes: potential trade-offs for the emotional and behavioral development of girls and boys. Psychol Bull, 132(1), 98-131.
Van Vugt, M., De Cremer, D., & Janssen, D. P. (2007). Gender differences in cooperation and competition: the male-warrior hypothesis. Psychol Sci, 18(1), 19-23.
Wagner, J. D., Flinn, M. V., & England, B. G. (2002). Hormonal response to competition among male coalitions. Evolution and Human Behavior, 23, 437-442.

---

## Round 0.2 · Minor Revisions

Dear Joyce

I am fully satisfied with your responses to the reviewers’ comments.
However, in addressing some of their concerns regarding the results, you appear to have inadvertently removed important information, specifically Table 1. That information, especially the statistical tests, seems vital to support this statement:

“As depicted in Table 1, at both ages more male than female focal children chose the play area containing the same-sex peer, but the sex difference was significant only at the older age.”

Therefore, please revisit and revise your Results section to ensure that you have presented all crucial information.

Sincerely,
Rob

---

## Round 0.3 · accepted · Accept

Sorry about the confusion over the Table Joyce. I think you were actually correct to upload it separately at this point. I apologize for the delay this caused.

More importantly, I'm glad this paper will now be published. It's another finding that further informs our understanding of sex differences and their developmental basis.